# Metagenomic Identification of Viral Sequences in Laboratory Reagents

**DOI:** 10.3390/v13112122

**Published:** 2021-10-21

**Authors:** Ashleigh F. Porter, Joanna Cobbin, Ci-Xiu Li, John-Sebastian Eden, Edward C. Holmes

**Affiliations:** 1The Peter Doherty Institute of Immunity and Infection, Department of Microbiology and Immunity, University of Melbourne, Melbourne, VIC 3000, Australia; ashleigh.porter@unimelb.edu.au; 2Sydney Institute for Infectious Diseases, School of Life and Environmental Sciences, The University of Sydney, Sydney, NSW 2006, Australia; joanna.cobbin@sydney.edu.au (J.C.); js.eden@sydney.edu.au (J.-S.E.); 3Sydney Institute for Infectious Diseases, School of Medical Sciences, The University of Sydney, Sydney, NSW 2006, Australia; 4Key Laboratory of Etiology and Epidemiology of Emerging Infectious Diseases in Universities of Shandong, Shandong First Medical University & Shandong Academy of Medical Sciences, Taian 271000, China; cixiu.li@sydney.edu.au; 5Centre for Virus Research, Westmead Institute for Medical Research, Westmead, NSW 2145, Australia

**Keywords:** reagent contamination, virology, metatranscriptomics, *Circoviridae*, *Totiviridae*, *Tombusviridae*, *Lentiviridae*

## Abstract

Metagenomic next-generation sequencing has transformed the discovery and diagnosis of infectious disease, with the power to characterise the complete ‘infectome’ (bacteria, viruses, fungi, parasites) of an individual host organism. However, the identification of novel pathogens has been complicated by widespread microbial contamination in commonly used laboratory reagents. Using total RNA sequencing (“metatranscriptomics”) we documented the presence of contaminant viral sequences in multiple ‘blank’ negative control sequencing libraries that comprise a sterile water and reagent mix. Accordingly, we identified 14 viral sequences in 7 negative control sequencing libraries. As in previous studies, several circular replication-associated protein encoding (CRESS) DNA virus-like sequences were recovered in the blank control libraries, as well as contaminating sequences from the *Totiviridae*, *Tombusviridae* and *Lentiviridae* families of RNA virus. These data suggest that viral contamination of common laboratory reagents is likely commonplace and can comprise a wide variety of viruses.

## 1. Introduction

Culture-independent methods, particularly metagenomic next-generation sequencing (mNGS), have revolutionised pathogen discovery, streamlined pathways of clinical diagnosis, and have enhanced our ability to track infectious disease outbreaks [1], including the current COVID-19 pandemic [2,3]. These methods can reveal the complete profile of pathogenic and commensal microorganisms within a host, comprising viruses, bacteria, fungi and eukaryotic parasites. As mNGS, particularly total RNA sequencing (i.e., metatranscriptomics), enables the identification of diverse and divergent viral sequences, it has been widely utilised for virus discovery [4,5,6,7,8].

Although the data generated by mNGS are bountiful and cost-effective, they come with several inherent limitations, central of which is the possibility of reagent contamination [9]. Indeed, the contamination of mNGS data can be problematic when identifying microbes in the context of disease association and creates issues when attempting to identify the true host of a novel microbe. The experimental preparation of samples for sequencing necessarily involves treatment with a variety of reagents, many of which have been shown to carry contaminating nucleic acids, including viral sequences [10,11,12,13,14,15]. Previous work has illuminated the extent of viral contamination in commonly used laboratory components, particularly those with small single-stranded (ss) DNA genomes [9,14,16,17,18]. Accordingly, there is a clear need for appropriate controls when characterising novel viruses from metagenomic data. For example, metagenomic analysis of human plasma samples revealed the presence of sequences of Kadipiro virus, a double-stranded positive-sense RNA virus [19], although these were later shown to likely reflect contamination [20]. An additional complication is that reagent-associated viral sequences are often not shared nor widespread across samples, only appearing intermittently [9].

Although mNGS has identified many novel viruses, diverse species of circular replication-associated protein encoding (CRESS) ssDNA viruses have been particularly prominent [21,22,23,24,25]. However, as noted above, ssDNA viruses, particularly CRESS viruses and their relatives including circoviruses, are common contaminants of reagents, leading to incorrect inferences on host associations [9,26]. As well as DNA viruses, a variety of other microbial sequences are present in laboratory reagents, including bacteria, RNA viruses, and eukaryotic parasites [9,20,27,28,29,30].

To further explore the diversity of contaminant sequences in laboratory components, particularly those derived from viruses, we used metatranscriptomics to investigate seven libraries of blank RNA sequencing samples that represent sterile water extractions and library preparation reagents.

## 2. Materials and Methods

### 2.1. Generating ‘Blank’ Sequencing Libraries

When generating total RNA sequencing libraries, we regularly utilise negative or ‘blank’ samples as experimental controls to assess the extent of reagent contamination. These controls are derived from extractions of the sterile water used at the elution step and, importantly, are expected to contain no nucleic acid material. In theory, these negative controls should generate no sequencing reads. However, they can capture contamination during the DNA/RNA extraction or library preparation steps. 

Herein, we analysed negative control sequencing libraries under different experimental conditions to identify likely contaminant sequences (Table 1). Total RNA was extracted using either the RNeasy Plus Universal Kit (Qiagen, Hilden, Germany), RNeasy Plus Mini Kit (Qiagen, Hilden, Germany) or the Total RNA purification Kit (Norgen BioTek Corp., Thorold, ON, Canada), as described in Table 1. RNA libraries were prepared with the Trio RNA-seq + UDI Library Preparation Kit (NuGEN) or the SMARTer Stranded Total RNA-Seq Kit v2-Pico Input Mammalian (Takara Bio, USA) and sequenced on the MiSeq, NextSeq or NovaSeq Illumina platforms, producing between 0.63 Gb and 8.7 Gb of data per library. 

### 2.2. Analysis of Virus-Like Sequences in Laboratory Reagents

Each sequencing library underwent trimming and the *de novo* assembly of reads, completed using the Trinity software with default settings [31] or MEGAHIT (L3) [32]. Sequence similarity searches using Diamond BLASTX were performed on the *de novo* assembled contigs against the GenBank non-redundant (nr) database [33,34]. Specifically, we used a combination of e-value, hit length, and percentage similarity to determine the potential of a contig to be a viral sequence. The abundance of reagent-associated reads was calculated by comparing the number of contig reads to the total number of library reads (via mapping trimmed reads back to the contigs) as performed in previous studies [5,8]. 

After initial identification, all potential contaminant sequences were subjected to phylogenetic analysis. To ensure high quality amino acid sequence alignments, only conserved sequence contigs that were >800 bp (>200 amino acids) in length were used in downstream analysis. Sequences of reference proteins, including the highly conserved replicase, DNA polymerase and RNA-dependent RNA polymerase (RdRp) protein domains, were downloaded from the NCBI RefSeq database (Table 2). Contig and reference proteins were aligned using the L-INS-I algorithm in MAFFT v7 [35], with ambiguously aligned regions removed using Gblocks [36]. This resulted in final sequence alignments of between 125 and 672 amino acids in length (Table 2). Phylogenetic trees of all alignments were then estimated using the maximum likelihood method in IQ-TREE [37], using the model testing option and bootstrap resampling with 500 replications. 

## 3. Results

In total, we identified 14 reagent-associated viral sequences in the negative (blank) control samples, including seven CRESS-like viral sequences, four novel *Tombusviridae*-like viral sequences, and single *Lentivirus*-like and *Totiviridae*-like viral sequences.

### 3.1. Abundance of Reagent-Associated Viral Reads

The abundance of reads in each library was calculated to compare the percentage of reads associated with viruses (Figure 1). This revealed that the virus-associated contigs identified were predominantly CRESS-like (Figure 1). The L5 library contained only one virus-associated contig, associated with *Escherichia coli* phage PhiX 174 DNA: this was intentionally added into the sequencing run to add complexity and improve signal in the library. Both the L4 and L6 libraries did not contain long (>800 bp) virus-associated contigs.

Novel reagent-associated virus-like sequences were identified in four of the seven libraries (Table 3). Seven novel circo-like viruses (termed Reagent-associated CRESS-like virus 1-7), four novel tombusvirus-like viruses (termed Reagent-associated tombus-like virus 1-4), and one totivirus-like and lentivirus-like sequence (termed Reagent-associated toti-like virus and Reagent-associated lenti-like virus, respectively) were identified in libraries L1, L2 and L3. The contigs ranged from 828 to 3878 bp in length and comprised 0.0001–1.27% of reads in their associated libraries.

### 3.2. Characterisation of Reagent-Associated Circovirus-Like and CRESS-Like Viruses

Due to extensive genetic diversity within the *Circoviridae* we inferred two separate sequence alignments and hence two phylogenetic trees, representing the (i) CRESS and CRESS-like viruses and (ii) circoviruses taken independently, although both were based on the Rep protein sequence (Figure 2 and Figure 3). All seven of the novel reagent-associated circovirus-like sequences exhibited greater sequence similarity to the CRESS viruses and therefore were included in the CRESS virus phylogeny. These were termed Reagent-associated CRESS-like viruses 1-7. These viruses occupied diverse locations across the phylogeny, although they were closely related to some previously identified reagent-associated viruses: Avon–Heathcote estuary associated circular viruses, *Circoviridae* sp. subtypes, Dromedary stool-associated circular virus subtypes, and Sandworm circovirus [5,9] (Figure 2). It is notable that the CRESS viruses analysed derive from a variety of environments, and there is no clear topological pattern according to the host species of sample origin, which is anticipated in the case of contaminant sequences. The seven novel CRESS-like viruses identified also varied in abundance in the L1 and L3 libraries (0.01–9.66%). In contrast, a phylogenetic analysis of the Rep protein of other members of *Circoviridae* (Table 2), containing what we hypothesise are bona fide viruses, reveals a pattern of host-based clustering (Figure 3). In particular, this phylogeny was characterised by two distinct clades of circoviruses: circoviruses, associated with vertebrate hosts, and cycloviruses associated with invertebrates.

### 3.3. Characterisation of Reagent-Associated Lentivirus-Like, Tombusvirus-Like and Totivirus-Like Sequences

Aside from ssDNA viruses, we identified an additional seven novel reagent-associated viral sequences in the blank control libraries. The first of these was a novel lentivirus-like sequence that we then used in an alignment of the retroviral Pol protein (Table 2). A phylogenetic tree was inferred from the alignment and the novel Reagent-associated lenti-like virus was shown to cluster closely with Equine infectious anaemia viruses (EIAV), although occupying a relatively long branch within this clade (Figure 4).

Similarly, we identified four novel tombus-like sequences in the blank control samples: these were termed Reagent-associated tombus-like virus 1-4. A sequence alignment of the RdRp protein domain was used to infer a phylogenetic tree of these tombusvirus-like sequences that are commonly associated with plants (Table 2). Three of the novel tombus-like viruses cluster together in the same divergent clade that falls basal to a majority of the tombus-like viruses (Figure 5). Only two previously described tombus-like virus sequences fall in more divergent positions—Wenzhou tombus-like virus 11 and Sclerotinia sclerotiorum umbra-like virus 1. As these were both identified in metatranscriptomic studies [8,38] it is possible that they reflect reagent contamination, although Sclerotinia sclerotiorum umbra-like virus 1 was found in two samples of *Sclerotinia sclerotiorum* (a fungus) compatible with its status as a true mycovirus [38,39]. Interestingly, Plasmopara viticola lesion associated tombus-like virus 2, also suggested to be a mycovirus, similarly falls into a relatively divergent clade (Figure 5). Finally, Reagent-associated tombus-like virus 3 was identified in blank library L3 at a relatively high abundance (>1% of total reads), although it had a shorter (1574 bp) and likely incomplete genome compared to most tombusviruses (~4–5 kb). This sequence falls basal to a clade within the broader tombusvirus tree that includes a variety of plant viruses, including Groundnut rosette virus, Carrot mottle virus and Tobacco mottle virus.

Finally, the remaining novel sequence was related to the totiviruses, a family of double-strand RNA viruses commonly associated with fungi. The novel totivirus-like sequence was termed Reagent-associated toti-like virus. It was used in an alignment of the RdRp protein domain (Table 2), from which a phylogenetic tree was estimated (Figure 6). This revealed that the sequence appears to be related to Scheffersomyces segobiensis virus (83% amino acid identity) associated with the fungus *Scheffersomyces segobiensis.*

## 4. Discussion

Viral sequences, particularly those with single-stranded DNA genomes, have previously been associated with common laboratory components [9], and these contaminant viral sequences have sometimes led to erroneous disease associations [14,17,18,20,40]. Herein, using a series of blank controls comprising sterile water and commonly used laboratory reagents, we identified a diverse range of viral sequences.

Few laboratory reagents appear to be entirely free from contamination, particularly by ssDNA viruses and predominantly circoviruses [5,9,26]. Indeed, approximately half of the viral sequences identified here were CRESS-like members of the *Circoviridae*. Unfortunately, high levels of sequence diversity prevented us from obtaining a meaningful alignment of the Rep protein for the novel CRESS-like virus sequences obtained here and known *Circoviridae*. Accordingly, we divided the family into sub-groups, denoted here as “host-associated circoviruses” (Figure 3) and “CRESS and CRESS-like viruses” and performed phylogenetic analyses on each (Figure 2). Notably, in the “host-associated circovirus” phylogeny viruses generally clustered according to the broad host species of origin. In contrast, within the CRESS and CRESS-like phylogeny, clades could not be defined based on specific hosts or environments, and while many samples were originally derived from marine- or faeces-associated environments, these sequences did not cluster together. Interestingly, however, one of viruses identified in this study, Reagent-associated CRESS-like virus 4, was most closely related to Avon–Heathcote Estuary associated circular virus 3, previously identified as a reagent-associated virus [41]. In addition, the seven novel CRESS-like sequences identified here were related to previously identified reagent-associated viruses, including those identified by Asplund et al. (highlighted in blue, Figure 2) [9], as well as Sandworm circovirus similarly proposed to be a reagent contaminant [42]. This strongly suggests that these sequences are likely associated with laboratory reagents. 

It is therefore clear that CRESS-like viruses are common experimental reagent contaminants, with widespread reagent-associated sequences dispersed throughout the CRESS phylogeny. This, along with the range of CRESS viruses of undetermined host origin, create major difficulties in determining the origin of novel CRESS viruses. Although there have been many new members of *Circoviridae* characterised in recent years, particularly cycloviruses [5,43,44], we suggest that current and future characterisations of novel circovirus- and CRESS-like genomes should be completed cautiously with additional confirmation steps. A review of the current members of *Circoviridae* is also advisable as these likely include viruses with contaminant origins but incorrectly assigned to a specific host organism. As a case in point, multiple cycloviruses are listed as having vertebrate hosts (such as Human cyclovirus VS5700009) even though these viruses are normally associated with invertebrates.

We also identified several tombusvirus-like sequences in this study, as well as a totivirus- and lentivirus-like sequence. The *Tombusviridae* are a family of single-strand positive-sense RNA viruses which are usually associated with mosaic diseases in plants. We identified four novel tombusvirus-like sequences associated with laboratory reagents, calling into question the provenance of other novel tombusviruses identified in some meta-transcriptomic studies [45]. The identification of reagent-associated tombusvirus-like sequences suggests that additional care should be taken when characterising novel tombusvirus sequences, particularly when associating novel or previously identified tombusviruses with a host or disease. Similarly, although the natural hosts of the *Totiviridae* are fungi, other *Totiviridae* are associated with human-infecting protozoa, such as *Trichomonasvirus* associated with *Trichomonas vaginalis* [46] and *Giardiavirus* that likely infects *Giardia lamblia* protozoa [47,48]. The novel Reagent-associated totivirus identified in this study is distantly related to known totiviruses. We recommend that caution be taken when identifying novel totiviruses, especially if they are related to reagent-associated toti-like virus. 

Lentiviruses are a genus within the *Retroviridae* and are well documented in a wide range of vertebrate species. The novel sequence identified in this study—reagent-associated lenti-like virus—is closely related to several known sequences of equine infectious anemia virus (EIAV) that cause the chronic disease, equine infectious anaemia (EIA) in horses. EIAV is transmissible through bodily secretions [49,50], and has been suggested to be vector-borne through biting flies [51]. Although the novel reagent-associated lenti-like virus was genetically distinct from known EIAV sequences, care should obviously be taken to ensure that any EIAV-like virus is a true viral infection rather than a reagent contaminant. 

## 5. Conclusions

This study further highlights the extent of viral sequences in commonly used laboratory reagents [9] and the power of mNGS to monitor contamination in microbiological laboratories [52]. Although the source of these contaminants is unknown and needs further scrutiny, we tentatively suggest that viral vectors (for example, in the *Lentiviridae*) represent a likely source. Factors to consider when assessing the presence of reagent contaminants include genome coverage, read depth and distribution of read alignments across genomes, and that potential contaminant sequences are often only present at low abundance and in multiple libraries. Importantly, reagent-associated viruses are often more prevalent in sequencing reads than assembled contigs, emphasising the importance of careful assessment when relying on read data alone for characterising novel viruses and other microbial genomes [9,26]. Interestingly, most viral reads identified were in L1-L3, all of which were generated using the RNeasy Plus Mini Kit (Qiagen, Hilden, Germany).

Finally, our work highlights the importance of employing additional steps such as PCR or cell culture to confirm the presence of the pathogen after initial metagenomic identification [9,26]. Clearly, sequencing negative controls, such as that using sterile water and reagent mix as performed here, should become normal procedure in quality control.

## Figures and Tables

**Figure 1 viruses-13-02122-f001:**
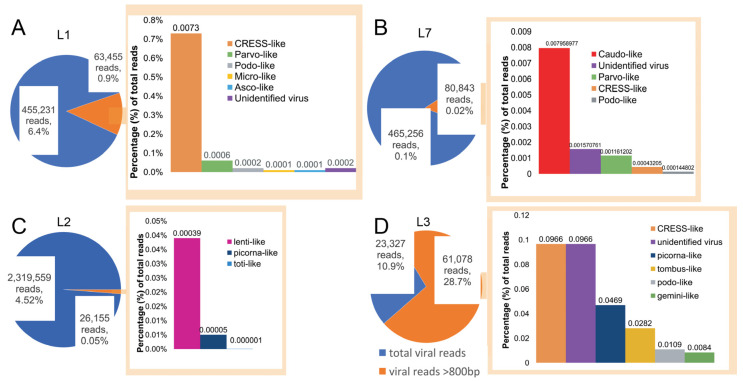
Abundance of viral reads in libraries L1, L2, L3, and L7. Visual representation of the virus-associated reads in respective libraries, with pie charts depicting the total number reads mapped to long (>800 bp) virus-associated contigs (orange) compared to all the virus-associated reads (blue). (**A**–**D**) Each bar chart denotes the proportion of contigs associated with different virus families in the respective libraries.

**Figure 2 viruses-13-02122-f002:**
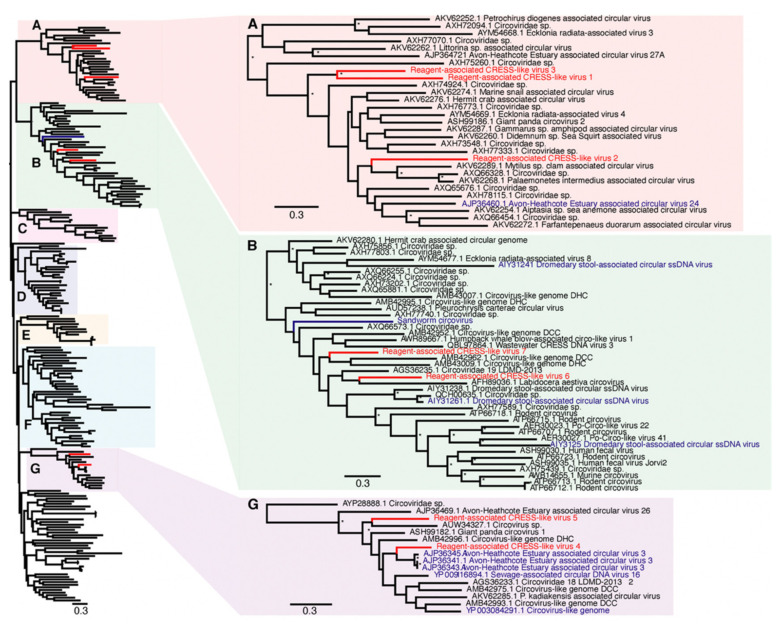
Phylogenetic relationships of CRESS (ssDNA) viruses, including the seven novel CRESS-like viruses identified here and highlighted in red (Reagent-associated CRESS-like viruses 1-7). Reagent-associated sequences determined previously are highlighted in blue. The clades that included the novel CRESS-like viruses identified here (**A**,**B**,**G**) are magnified on the right. The tree and other clades (**C**–**F**) are shown in higher resolution in Appendix A. The tree was mid-point rooted for clarity purposes only. Bootstrap values greater than 70% are represented by asterisks next to nodes. All horizontal branch lengths are scaled according to the number of amino acid substitutions per site.

**Figure 3 viruses-13-02122-f003:**
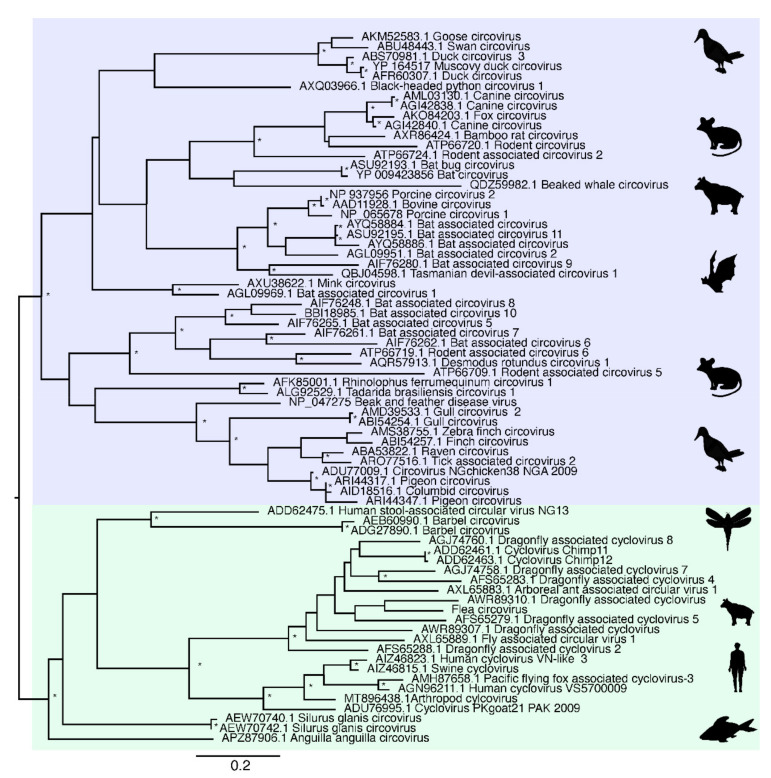
Phylogenetic relationships of the ssDNA virus family *Circoviridae* based on hypothesised “host-associated” circoviruses. The tree has two major clades, comprising the circovirus clade associated with vertebrate hosts (highlighted in blue) and the cyclovirus clade previously associated with invertebrate hosts (highlighted in green). For clarity, the tree is mid-point rooted. Bootstrap values greater than 70% are represented by asterisks next to nodes. All horizontal branch lengths are scaled according to number of amino acid substitutions per site.

**Figure 4 viruses-13-02122-f004:**
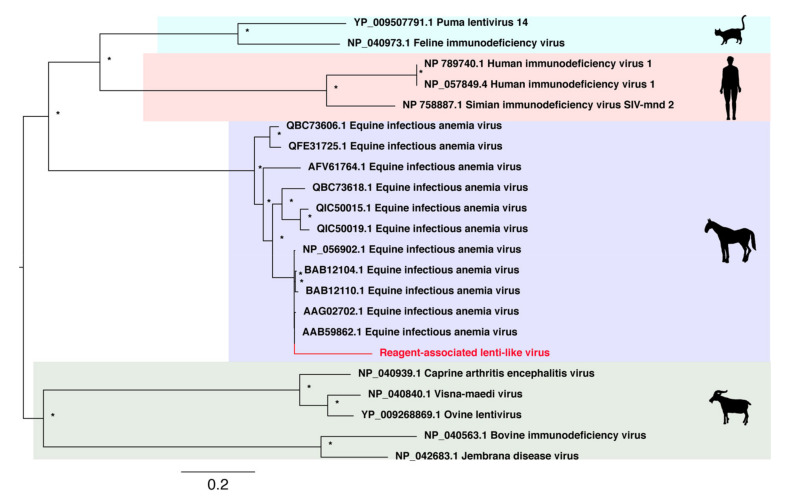
Phylogenetic relationships of RNA virus family *Lentiviridae* including the novel virus reagent-associated lenti-like virus sequence identified in this study. This virus is highlighted in red and falls within the EIAV clade.

**Figure 5 viruses-13-02122-f005:**
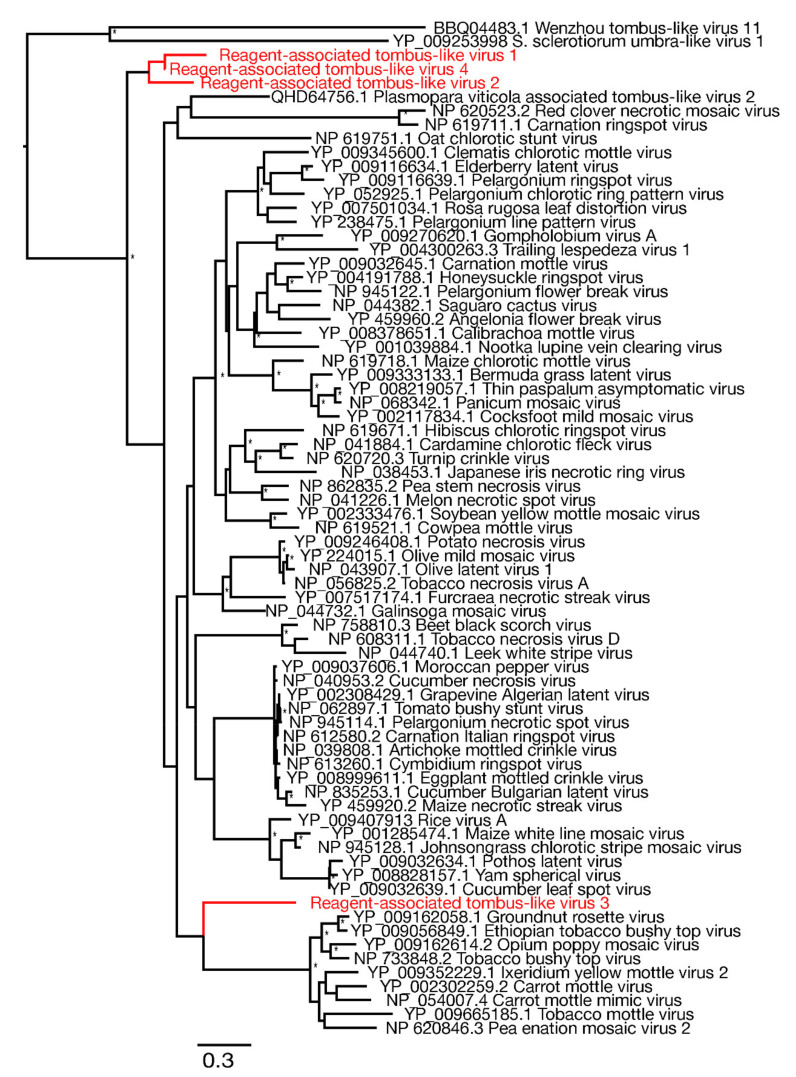
Phylogenetic relationships of RNA virus family *Tombusviridae* including the seven novel viruses identified in this study (highlighted in red). The phylogeny was mid-point rooted for clarity purposes only. Bootstrap values greater than 70% are represented by asterisks next to nodes. All horizontal branch lengths are scaled according to number of amino acid substitutions per site.

**Figure 6 viruses-13-02122-f006:**
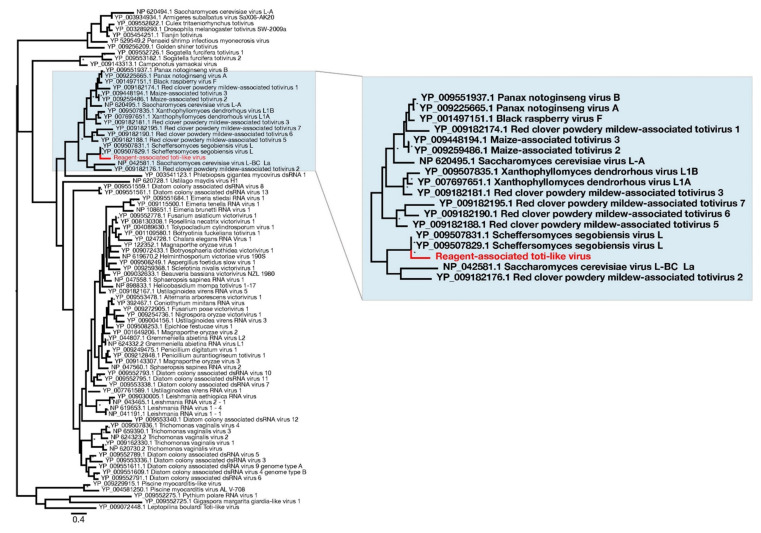
Phylogenetic relationships of RNA virus family *Totiviridae*, including the novel virus identified in this study—Reagent-associated toti-like virus (highlighted in red). For clarity, the tree was mid-point rooted. Bootstrap values greater than 70% are represented by asterisks next to nodes. All horizontal branch lengths are scaled according to number of amino acid substitutions per site.

**Table 1 viruses-13-02122-t001:** Experimental conditions of each blank negative control sample utilised here.

Library Name	Sequencing Platform	RNA Extraction	Library Preparation	Data Generated	SRA Library Accession
L1	Illumina Novaseq 6000 150 cycle kit (2 × 75 nt reads)	RNeasy Plus Universal Kits (Qiagen, Hilden, Germany)	Trio RNA-seq + UDI (NuGEN)	11,940,824 paired reads (1.8 Gb)	SRR14737471
L2	Illumina Novaseq 6000 150 cycle kit (2 × 75 nt reads)	RNeasy Plus Universal Kits (Qiagen, Hilden, Germany)	Trio RNA-seq + UDI (NuGEN)	57,606,392 paired reads (8.7 Gb)	SRR14737470
L3	Illumina MiSeq, v3 150 cycle kit (2 × 75 nt reads)	RNeasy Plus Mini Kit (Qiagen, Hilden, Germany)	SMARTer Stranded Total RNA-Seq Kit v2 -Pico Input Mammalian (Clontech)	4,156,504 paired reads (0.63 Gb)	SRR10069984
L4	Illumina NextSeq 500, mid-output 150 cycle kit (2 × 75 nt reads)	Total RNA Purification Kit (Norgen Biotek, Thorold, ON, Canada)	SMARTer Stranded Total RNA-Seq Kit v2-Pico Input Mammalian (Clontech)	32,279,914 paired reads (4.91 Gb)	SRR14737469
L5	Illumina MiSeq 150 cycle kit (2 × 75 nt reads)	Total RNA purification Kit (Norgen BioTek Corp., Thorold, ON, Canada)	SMARTer Stranded Total RNA-Seq Kit v2-Pico Input Mammalian (Clontech)	7,342,876 paired reads (1.10 Gb)	SRR15221433
L6	Illumina MiSeq 150 cycle kit (2 × 75 nt reads)	Total RNA purification Kit (Norgen BioTek Corp., Thorold, ON, Canada)	SMARTer Stranded Total RNA-Seq Kit v2-Pico Input Mammalian (Clontech)	10,978,253 paired reads (1.65 Gb)	SRR15221432
L7	Illumina MiSeq 150 cycle kit (2 × 75 nt reads)	Total RNA purification Kit (Norgen BioTek Corp., Thorold, ON, Canada)	SMARTer Stranded Total RNA-Seq Kit v2-Pico Input Mammalian (Clontech)	8,564,269 1.28 Gb	SRR14737466

**Table 2 viruses-13-02122-t002:** Reference proteins for each sequence alignment performed in this analysis.

Reference Protein	Protein Acronym	Virus Taxonomy	Number of Sequences in Analysis	Alignment Length (Amino Acid, AA)
Viral replicase protein	Rep	CRESS	221	672 AA
Viral replicase protein	Rep	*Circoviridae*	69	161 AA
Polymerase peptide	Pol	*Lentiviridae*	11	478 AA
RNA-dependent RNA polymerase	RdRp	*Totiviridae*	95	125 AA
RNA-dependent RNA polymerase	RdRp	*Tombusviridae*	87	256 AA

**Table 3 viruses-13-02122-t003:** Novel reagent-associated viral sequences identified in this study.

Virus name	Accession	Library Abundance (%) of Total Reads (rRNA Removed)	Length (bp)	Library
Reagent-associated tombus-like virus 1	MZ824229	1.28	1204	L3
Reagent-associated tombus-like virus 2	MZ824228	0.46	828	L3
Reagent-associated tombus-like virus 3	MZ824227	1.08	1574	L3
Reagent-associated tombus-like virus 4	MZ824226	1.29	1410	L3
Reagent-associated toti-like virus	MZ824225	0.001	920	L2
Reagent-associated lenti-like virus	MZ824230	0.004	962	L2
Reagent-associated CRESS-like virus 1	MZ824237	0.78	3878	L1
Reagent-associated CRESS-like virus 2	MZ824236	0.24	2377	L1
Reagent-associated CRESS-like virus 3	MZ824235	0.02	1592	L1
Reagent-associated CRESS-like virus 4	MZ824234	2.89	2663	L3
Reagent-associated CRESS-like virus 5	MZ824233	9.66	3027	L3
Reagent-associated CRESS-like virus 6	MZ824232	4.98	3517	L3
Reagent-associated CRESS-like virus 7	MZ824231	0.01	1124	L1

## Data Availability

The viral genome sequence data generated in this study have been deposited in the NCBI database under accession numbers MZ824225-37. Sequence reads are available at the public Sequence Read Archive (SRA) database with accession SRX6803604 and under the BioProject accession PRJNA735051 reference numbers SRR14737466-71 and BioSample numbers SAMN20355437-40.

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
