# Peer review of "Metagenomic Identification of Viral Sequences in Laboratory Reagents"

_viruses, 2021, doi:10.3390/v13112122_

Round 1

Reviewer 1 Report

In the manuscript entitled “Metagenomic identification of viral sequences in laboratory reagents” Ashley Porter et al. describe the generation of “blank” RNA sequencing libraries in order to estimate the risk of recovering viral sequences in NGS data that originate from contamination during the wet-lab part of sample processing. To this end, they took the “sterile” water provided as eluent in RNA extraction kits and used it as blank input for the subsequent cDNA library generation. After de novo assembly of the primary sequence reads, the authors identified contigs of viral origin via BLASTX homology search in the GenBank non-redundant (nr) database. In this way, the authors identified 13 “reagent-associated” novel viruses for which they were able to retrieve high quality contigs >800bp in length. This included seven cases of CRESS-DNA viruses, four tombus-like viruses, one toti-like and one lenti-like virus. The provided phylogenetic analyses support the notion that specifically CRESS-DNA viruses found in metagenomic/-transcriptomic data frequently represent contaminants of laboratory material, which is a known problem that gained high attention in the scientific community over the recent years.

The provided work points out the importance of a critical interpretation of NGS data dedicated to virus discovery. The main take-home message is that one should include blank controls when conducting such studies.

Comments:

  • Commonly, viral contaminations in NGS data are attributed to silica spin columns for nucleic acid extraction (Ref. 14, 17, 41). This study indicates that the source of viral contamination is either the water (not the columns) of the extraction kits or any downstream factor such as the components of the library preparation kits or the sequencing devices. The authors should point this out in a bit more detail in the discussion.
  • Do the authors know what kind of “sterile” water is provided in the nucleic acid extraction kits? Is it just “ultrapure” (=dirty) deionized, 0.45µm-filtered and autoclaved water or double-distilled water? A related request to the technical customer service of the kit manufacturers would be very welcome. If they don’t answer or decline to provide this information, this should be briefly mentioned in the discussion. Note: In our department we only use medical-grade double-distilled water (intended for injections) for lab procedures prone to contamination.
  • The presence of CRESS-DNA viral sequences in the RNAseq libraries deserves further discussion. Do the authors believe that this results from contamination with viral genomic DNA or from reverse transcription of viral mRNA transcripts? How could one discriminate between both possibilities?
  • Can the authors provide any (gross) estimate of the number of contaminant viral particles/genomes that would lead to the observed results? Are there any differences in read coverage/sequencing depth between the different kinds of viruses (RNA vs. DNA viruses, linear vs. circular genomes, coding vs. non-coding genomic regions)? Is it possible to derive any kind of read coverage threshold, below which it is likely a contaminant, above which it is likely a true infection of the sampled material?
  • Is there any significant difference in contaminant frequency/abundancy between the different RNA extraction kits, library preparation kits, sequencing platforms?

Author Response

Reviewer #1

  • Commonly, viral contaminations in NGS data are attributed to silica spin columns for nucleic acid extraction (Ref. 14, 17, 41). This study indicates that the source of viral contamination is either the water (not the columns) of the extraction kits or any downstream factor such as the components of the library preparation kits or the sequencing devices. The authors should point this out in a bit more detail in the discussion.

Response: We thank the reviewer for this comment. As we were unable to pinpoint exactly where the viral reads come from – this is rather beyond the scope of this study – we did not specifically exclude the spin columns as a potential source of contamination. Future studies will hopefully address this.

  • Do the authors know what kind of “sterile” water is provided in the nucleic acid extraction kits? Is it just “ultrapure” (=dirty) deionized, 0.45µm-filtered and autoclaved water or double-distilled water? A related request to the technical customer service of the kit manufacturers would be very welcome. If they don’t answer or decline to provide this information, this should be briefly mentioned in the discussion. Note: In our department we only use medical-grade double-distilled water (intended for injections) for lab procedures prone to contamination.

Response: Using medical-grade water for contamination-prone work is good idea and something we will consider this for future work. RNAse free water is commonly supplied in RNA extraction kits.

  • The presence of CRESS-DNA viral sequences in the RNAseq libraries deserves further discussion. Do the authors believe that this results from contamination with viral genomic DNA or from reverse transcription of viral mRNA transcripts? How could one discriminate between both possibilities?

Response: Because we are using RNA-Seq we are (in theory) able detect any organism that expresses RNA. Obviously, even though they have DNA genomes, CRESS viruses also fall into this category. With respect to CRESS virus contamination introduced by laboratory reagents, it would all depend on when the DNAse step was used in the process. To resolve this requires a more complex investigation that is unfortunately beyond the scope of the current study.

  • Can the authors provide any (gross) estimate of the number of contaminant viral particles/genomes that would lead to the observed results? Are there any differences in read coverage/sequencing depth between the different kinds of viruses (RNA vs. DNA viruses, linear vs. circular genomes, coding vs. non-coding genomic regions)? Is it possible to derive any kind of read coverage threshold, below which it is likely a contaminant, above which it is likely a true infection of the sampled material?

Response: This is a very important but also very difficult question to resolve. Unfortunately, we currently do not have a simple way to answer this question and hence would rather not speculate. However, we are looking forward to future studies that will focus on this concept and will hopefully provide more clarity about the expected read coverage of “true” versus contaminant viruses.

  • Is there any significant difference in contaminant frequency/abundancy between the different RNA extraction kits, library preparation kits, sequencing platforms?

Response: The reviewer raises an important point. We have added a line in the discussion to resolve this.

Reviewer 2 Report

Porter et al. present an analysis of viral sequences identified in “blank” metatranscriptomes, i.e. likely deriving from reagent contamination. This type of analysis is extremely important to perform and share with the research community as metatranscriptomics is increasingly used for virus discovery in clinical/epidemiological contexts. The current manuscript is extremely clear and provides not only new examples of likely reagent contaminants (which the authors contributed to public databases), but also a framework for other groups to perform similar analysis of their own negative controls. Hence, I only have a few minor remarks and suggestions for the authors.

l. 18: “multiple libraries of ‘blank’ negative control sequencing libraries” seems a bit unclear, maybe “multiple ‘blank’ negative control sequencing libraries” or “multiple libraries of ‘blank’ negative control samples” ?

l. 29: While it is a matter of personal preference, I favor and would suggest “high-throughput sequencing” (HTS) rather than “next-generation sequencing” (NGS).

l. 82: Were Trinity and Megahit both used for each library, or was a single assembly (Trinty or Megahit) performed per library ? If the latter, please include the assembler used for each library in Table 1.

l. 85: “used a combination of e-value, hit length, and percentage similarity”: If possible, please specify which specific cutoffs were used for e-value, hit length, and percentage of similarity.

l. 114: should “total number of long (>800 bp) virus-associated contigs (orange)” be “total number of reads mapped to long (>800 bp) virus-associated contigs (orange)” ?

l. 116: “contigs of associated” should be “contigs associated”

l. 144: “cycloviruses associated with invertebrates.”: Based on Fig. 3, a number of cycloviruses are currently linked to vertebrate “hosts” (e.g. Cylovirus Chimp11 and Chimp12, Human cyclovirus VN-like 3, etc). Can the authors maybe add one sentence to comment on these sequences, i.e. do they believe that cycloviruses hosts span across vertebrates and invertebrates, or is there another explanation (e.g. virus present in a “human” sample but not infecting a human host ?).

l. 178: “to majority” should be “to a majority”

Table 1: If possible, please harmonize the identifiers used for library accession (right now, this column includes SRA experiment IDs, BioSample, or SRA run IDs for different libraries).

Author Response

Reviewer #2

  • 18: “multiple libraries of ‘blank’ negative control sequencing libraries” seems a bit unclear, maybe “multiple ‘blank’ negative control sequencing libraries” or “multiple libraries of ‘blank’ negative control samples” ?

Response: We thank the reviewer for notifying us. This has been corrected.

  • 29: While it is a matter of personal preference, I favor and would suggest “high-throughput sequencing” (HTS) rather than “next-generation sequencing” (NGS).

Response: We understand the point made, but prefer next-generation sequencing as we believe that term is more commonly used. It is, however, merely a matter of taste.

  • 82: Were Trinity and Megahit both used for each library, or was a single assembly (Trinty or Megahit) performed per library ? If the latter, please include the assembler used for each library in Table 1.

Response: Corrected.

  • 85: “used a combination of e-value, hit length, and percentage similarity”: If possible, please specify which specific cutoffs were used for e-value, hit length, and percentage of similarity.

Response: The specific cut-offs have been outlined in detail in the methodology sections of our previous papers. It therefore seems unnecessary to repeat it here.

  • 114: should “total number of long (>800 bp) virus-associated contigs (orange)” be “total number of reads mapped to long (>800 bp) virus-associated contigs (orange)” ?

Response: Corrected.

  • 116: “contigs of associated” should be “contigs associated”

Response: Corrected.

  • 144: “cycloviruses associated with invertebrates.”: Based on Fig. 3, a number of cycloviruses are currently linked to vertebrate “hosts” (e.g. Cylovirus Chimp11 and Chimp12, Human cyclovirus VN-like 3, etc). Can the authors maybe add one sentence to comment on these sequences, i.e. do they believe that cycloviruses hosts span across vertebrates and invertebrates, or is there another explanation (e.g. virus present in a “human” sample but not infecting a human host ?).

Response: We thank the reviewer for raising this. Good point. We have now added a relevant sentence to the Discussion section of our paper.

  • 178: “to majority” should be “to a majority”

Response: Corrected.

  • Table 1: If possible, please harmonize the identifiers used for library accession (right now, this column includes SRA experiment IDs, BioSample, or SRA run IDs for different libraries).

Response: Fair point. Now corrected.